# Quantitative Measures of Physical Risk Factors Associated with Work-Related Musculoskeletal Disorders of the Elbow: A Systematic Review

**DOI:** 10.3390/ijerph16010130

**Published:** 2019-01-05

**Authors:** David H. Seidel, Dirk M. Ditchen, Ulrike M. Hoehne-Hückstädt, Monika A. Rieger, Benjamin Steinhilber

**Affiliations:** 1University Hospital Tuebingen, Institute of Occupational and Social Medicine and Health Services Research (IASV), 72074 Tuebingen, Germany; Monika.Rieger@med.uni-tuebingen.de (M.A.R.); Benjamin.Steinhilber@med.uni-tuebingen.de (B.S.); 2Institute for Occupational Safety and Health of the German Social Accident Insurance (IFA), Unit Ergonomics, Referat Ergonomie, 53757 Sankt Augustin, Germany; Dirk.Ditchen@dguv.de (D.M.D.); ulrike.hh63@gmail.com (U.M.H.-H.)

**Keywords:** elbow disorders, epicondylitis, ulnar neuropathy, quantitative measures, physical risk factors, work-related, force, repetition, posture

## Abstract

Background: Work-related musculoskeletal disorders at the elbow are a common health problem, which highly impacts workers’ well-being and performance. Besides existing qualitative information, there is a clear lack of quantitative information of physical risk factors associated with specific disorders at the elbow (SDEs). Objective: To provide evidence-based quantitative measures of physical risk factors associated with SDEs. Methods: Studies were searched from 2007 to 2017 in Medline, EMBASE, and Cochrane Work. The identified risk factors were grouped in main- and sub-categories of exposure using the Grading of Recommendations, Assessment, Development and Evaluation (GRADE) framework for rating evidence. Results: 133 different risk-factor specifications were identified in 10/524 articles and were grouped into 5 main- and 16 sub-categories of exposure. The risk factors were significantly associated with lateral epicondylitis, medial epicondylitis, or ulnar neuropathy. Significant risk factors such as wrist angular velocity (5°/s, with increasing prevalence ratio of 0.10%/(°/s), or forearm supination (≥45° and ≥5% of time combined with forceful lifting) were found. Conclusions: This review delivers a categorization of work-related physical risk-factor specifications for SDEs with a special focus on quantitative measures, ranked for evidence. These results may build the base for developing risk assessment methods and prospective preventive measures.

## 1. Introduction

Work-related musculoskeletal disorders are a common problem with great effects on workers’ health and the global economy. Work-related upper limb disorders (WRULDs) account for 20% to 45% [1,2] of all work-related musculoskeletal disorders. Within the WRULDs, elbow diseases significantly impact workers health by accounting for approximately 20% of those occupational diseases [2]. According to the European Agency for Safety and Health at Work [2], the costs of WRULDs are estimated between 0.5% and 2.0% of the Gross National Product. In Great Britain, 3.9 million working days were lost due to WRULDs in 2016/2017 [1], which illustrates the great need for preventive measures.

The first step in developing adequate preventive measures is to identify work-related risk factors. Associations between WRULDs and physical risk factors have been reported for years [3,4,5]. Besides the hand, wrist, and shoulder, the elbow also seems to be affected by physical exposure arising from different occupational activities [6]. For instance, two literature reviews refer to repetitive movements, awkward hand and forearm postures, and external force interactions as possible risk factors for specific disorders at the elbow (SDEs) [3,4]. The focus of these reviews was to present qualitative information about the relationship between physical risk factors and SDEs, and thus, to identify targets of preventive measures.

To develop adequate risk assessment tools, more detailed information about relevant risk factors and their quantitative specifications are required. In this regard, the systematic review by van Rijn et al. [6] presents some quantitative exposure-response relationships between work-related factors and SDEs between 1966 and 2007 [6]. The authors found associations between four work-related SDEs (in descending order according to prevalence: lateral epicondylitis (LE), medial epicondylitis (ME), cubital tunnel syndrome, and radial tunnel syndrome) and certain risk-factor specifications at work. Besides psychosocial factors, they identified physical risk-factor specifications, such as handling tools >1 kg (odds ratio (OR) 2.10 to 3.00), handling loads >5 kg (2 times/min for more than 2 h per day (h/day)), high hand grip forces for >1 h/day (OR 2.20 to 2.60), repetitive movements >2 h/day over 9 to 19 or ≥20 years (OR 2.20 to 3.60), arm lifting or hand bending for more than 25 or 75% of working time (OR 2.00 to 7.40), or working with vibrating tools for >2 h/day (OR 2.20 to 2.90) [6].

Over the last decade, great advancements have been made in the application of technical devices to measure physical exposure at the workplace [7,8,9,10], and current knowledge about relevant risk factors for SDEs has grown significantly.

Therefore, we augmented the work of van Rijn et al. [6] and conducted a systematic review on work-related physical risk factors for SDEs between 2007 and 2017, focusing on quantitative measures. We predominantly focused on the same four diseases reported by van Rijn et al. [6]. However, we were also open to other relevant SDEs.

The aim was to deliver a valid source of reference values for preventive purposes and, in particular, for developing adequate risk assessment methods at the workplace.

## 2. Materials and Methods

### 2.1. Literature Search and Selection Process

Our systematic procedure followed the item checklist for creating a systematic review by the Preferred Reporting Items for Systematic reviews and Meta-Analyses (PRISMA) statement [11,12] (Appendix A).

One author (D.H.S.) performed the literature search using the databases MEDLINE, EMBASE, and Cochrane Work from September 2007 to February 2017. Additionally, reference lists and peer-reviewed grey literature were scanned manually. Our search strategy was developed a priori and was based on keywords, Medical Subject Headings (MeSH) terms, free texts, and shortcuts for SDEs also used in previously published reviews [4,6,11,12,13,14,15,16,17,18]. Relevant keywords were applied, such as elbow pathologies, epicondylitis (lateral, medial), cubital tunnel syndrome, radial tunnel syndrome, pronator teres syndrome, tendinopathy, tenosynovitis, tendovaginitis, occupational exposure, and risk factor. The complete search strategy, including all keywords, is available as Appendix A (Appendix A).

To select eligible publications, we followed the 4 steps of the PRISMA Flow Diagram (1: Identification, 2: Screening, 3: Eligibility, and 4: Included Articles [11,12], see Figure 1). D.H.S. performed steps 1 and 3, and steps 2 and 4 were performed independently and blinded by D.H.S. and B.S. In case of disagreement, decisions regarding inclusion, exclusion, or methodological quality scores were achieved through discussion or by consulting a third author (D.M.D.).

### 2.2. Inclusion Criteria

All inclusion criteria were defined a priori. That means: Studies that did not meet the scope, i.e., studies without pathologies at the elbow or information about physical exposures, animal studies, and human analgesic studies were sorted out in advance. We considered the review by van Rijn et al. [6] as our key paper and supplemented their inclusion criteria as follows:(a)English or German as publication language, abstract available, peer-reviewed, no case studies/case reports, published after 01 September 2007, no acute traumata or bone fractures;(b)working adults as population (aged from 18 to 65);(c)quantitative or semi-quantitative (ordinal scaled) descriptions of exposure measures;(d)main study outcome of at least one SDE: LE or ME, cubital tunnel syndrome, radial tunnel syndrome, ulnar nerve entrapment, median nerve entrapment, pronator teres syndrome, tenosynovitis, tendovaginitis;(e)association between work-related physical risk factors in quantitative measures and at least one SDE.

The status of all articles concerning their inclusion is available in Appendix A (Appendix A). The PICO criteria (Population, Intervention, Comparison, Outcome) [11] were checked during steps 2 and 3.

### 2.3. Quality Assessment

A priori, we composed a criteria list for general assessment of the methodological quality based on the one by van Rijn et al. [6], and augmented it with additional items according to Padula et al. [19] and Sanderson et al. [20]. The total list contains 18 items in 5 categories and 3 score decisions (available (+), not available (−), or unclear (?), see Table 1). This assessment tool was eligible for different study designs such as cohort studies (CH), cross-sectional studies (CSS) or case-referent studies (CRS).

According to Padula et al. [19] and Wong et al. [21], the levels of general methodological quality were classified into 3 categories:high (high frequency of positive values ‘+’ ≥67% corresponds to a score ≥12),medium (medium frequency of positive values ‘+’ 66% to 34% corresponds to a score 6 < 12), andlow (low frequency of positive values ‘+’ ≤33% corresponds to a score ≤6).

With our focus on a more detailed grading of exposure and outcome assessment, we also chose the scoring system by Sulsky et al. [22] and adapted their assessment method for the hip joint to the elbow joint (see Table 2).

### 2.4. Level of Evidence and Data Analysis

References (author, date), study characteristics (design, samples/population attributes), outcomes, and relevant physical risk factors (definitions, declarations, exposure assessments, measures with corresponding 95%-confidence intervals (CIs)), sex/specific information, outcome assessments, and confounder adjustments) were extracted from the included original articles (via text, tables, and graphics [23]) by D.H.S. Risk-factor specifications will be reported as given in the original studies. If the data are presented as adjusted (e.g., for gender by multivariate statistical analyses) and unadjusted (univariate statistical analyses) findings, then only the adjusted results will be reported. We further reported all kind of relative risk indicator measures such as ORs, hazard ratios, or prevalence ratios.

For easier reading, potential risk factors will be divided in statistically significant and non-significant results based on the *p*-value (<0.05) and the lower 95%-CI limit (>1.0) [24].

Single results were gathered in main- and sub-categories of exposure, similar to Melhorn et al. [14]. Following van Rijn et al. [6], the attributed scores of the quality assessment, as well as evidence levels were taken into account for data interpretation. To assess the validity and evidence of the results, we used an established method in systematic reviews (GRADE—Grading of Recommendations Assessment, Development and Evaluation [25]): All evidence judgments and bias assessments for potential risk factors were achieved for sub-categories of exposure by applying the special GRADE framework for prognostic factor research [26,27,28,29].

A calculation of the results in the form of a meta-analysis would be considered only for very homogeneous designs of the included studies.

## 3. Results

### 3.1. Included Studies

From 524 identified articles, including 30 duplicates, 494 articles were scanned via title and abstract. After that, 322 articles were excluded with an initial agreement of 84.21% between the two authors D.H.S. and B.S., and a moderate interrater agreement (Kappa = 0.60 [30]). 141 of the remaining 172 articles were excluded after full text screening, and another 21 were excluded during eligibility assessment.

Studies were typically excluded due to a lack of quantitative measures of risk factors and the indication of diffuse elbow disorders without clear diagnosis (e.g., complaints or pain). Results of two systematic reviews [4,6] and one meta-analysis [5] which were already presented in included primary studies, were not listed additionally.

The selection of the studies is shown in Figure 1. Finally, 10 studies met our inclusion criteria and were used for further procedures such as assessing the study quality or extracting risk-factor specifications.

### 3.2. Quality of the Included Studies

#### 3.2.1. Methodological Quality

Five relevant cross-sectional studies (1 with high [31], 4 with medium quality [32,33,34,35]) were identified. Furthermore, 3 high-quality cohort studies [36,37,38], one high-quality triple case-referent study [39] and one medium quality case-referent study [40] were included. Overall quality of the included studies was rated as high. As the studies showed very different designs, pooling for a meta-analysis did not seem to be reasonable. In general, the main weaknesses in the designs of the included studies were low participation rates, unclear definitions of exposed and unexposed groups, lack of information about blinding status of the examiners (exposure and outcome), and minor reported statistical methods. 

#### 3.2.2. Quality of Exposure and Outcome Assessment

The exposure assessment showed scores of “1” (*n* = 1 [39]), “3” (*n* = 6 [32,33,35,36,38,40]), and “5” (*n* = 3 [31,34,37]), while the outcome assessment was scored with the highest possible score (“3”) in all included studies. In other words, all studies chose at least physical examinations to investigate the outcome, but only three of them [31,34,37] chose measurements for the assessment of exposure. The results are presented in Table 3 (in order of decreasing score for general methodological quality).

### 3.3. Physical Risk Factors Associated with SDEs

Only half of the included studies [31,33,34,37,39] described the risk-factor specifications in detail, providing clear definitions of these factors and giving additional further information on the examined exposures. All risk factors (including information on study design, subject groups, exposure and outcome determination and further study attributes) were listed in the Appendix A (Appendix A) in alphabetical order of the authors.

All included studies provided a total of 133 different risk-factor specifications (numbered from #1 to #133). Of these, 3 specifications (#26 [37]; #9 [38]; #42 [38]) were reported twice, first as the results of a cross-sectional study and then in a subsequent longitudinal cohort study. Dividing all specifications in statistically significant and non-significant associations led to 44 different significant associations (#1 to #44) and to 89 different non-significant (#45 to #133) associations. The significant associations are shown in Table 4 and in most cases the statistical analyses were adjusted for at least one confounder such as age or gender (*n* = 32). Twelve of the significant risk-factor specifications indicate a dose-response relationship when pairs for the same exposure were compiled (#3 and #4 [39]; #13 and #14 [36]; #16 and #17 [34]; #29 and #30 [31]; #31 and #32 [31]; #41 and #42 [32]). In 14 specifications both significant and non-significant associations between risk factors and specific SDEs were found (#2, #9, #10, #12, #13, #16, #17, #19, #22, #26, #40, #41, #42, and #43). All non-significant associations are listed in Appendix A (Appendix A). These were predominantly based on univariate analyses (60 out of 89 non-significant associations).

Significant associations between physical exposures and SDEs have only been verified for LE (9 studies [31,32,33,34,35,36,37,38,40]), ME (4 studies [33,34,35,36]), and for ulnar neuropathy (UN, 1 study [39]). Non-significant associations for these could be found as well (LE, 8 studies [31,32,33,34,36,37,38,40]; ME, 3 studies [33,34,36]; UN, 1 study [39]) while one study [33] also reported on two other diseases (radial tunnel syndrome, pronator teres syndrome) but with non-significant associations. All associated physical risk factors can be categorized into 5 main exposure groups (force, repetition, posture/movement, vibration, and combined exposures) and 16 sub-categories of exposure. The distribution of all risk-factor specifications among the 5 main exposure groups (inclusive numbers of different exposure determinations, numbers of referring studies) were presented in Figure 2 (for more details about risk-factor specifications and exposure categories: see Table 4, Appendix A (Appendix A), Appendix A (Appendix A)).

#### 3.3.1. Force

Risk factors #3 and #4 (forceful work with <10 or 10 to 29% of maximal strength, defined by [41]) were significantly associated with UN (OR 2.73 to 3.85) [39]. For this, a dose-response relationship was presented with multivariate analysis.

Hand in forceful grip ≥4 h/day was described as a significant risk factor both for ME (OR 3.80) and for LE and ME (OR 2.80) [36]. Less than 4 h (#46, #47) showed no significant associations. All these results were supported by univariate analyses.

Spahn et al. [40] found high ORs for maximum force >1 h/day (#1) for LE in men (OR 6.90) and in women (OR 9.60). Moreover, patting with the hand for >1 h/day (#5) was highly associated with LE in men (OR 13.80). These two specifications were supported by sex-adjusted calculations. On the other hand, constant moving or lifting or carrying of loads (#51, #52) were not significantly associated with LE.

Forceful lifting >0% of time and forceful lifting (≥2 times/min) were identified as two further significant risk-factor specifications for LE (OR 2.65 to 3.06) [31]. These results were adjusted for age (continuously), gender and BMI (continuously). Fan and colleagues [31,37] defined forceful exertion as pinch grip force ≥8.9 N or power grip force ≥44.1 N and forceful lifting with an object weight of 0.9 kg (pinch grip) or 4.5 kg (power grip), considering other studies [42,43,44]. Lifting ≥3% time (#50 [37]) or less than 2 times/min (#53 [31]) was not significant for LE, in contrast. Furthermore, non-significant results could be found for different levels of muscular activity (#48, #49) via sex-adjusted analysis [34].

#### 3.3.2. Repetition

More than 4 wrist or elbow movements per minute ≥2.5 h/day (#11) were significantly associated with UN (OR 2.22) by using partly adjusted models [39]. If the exposure time was less than 2 h/day (#57), associations between risk factors and elbow disorders were not found to statistically significant.

Doing repetitive tasks >4 h/day (#9) was only identified as a risk factor for LE in women (OR 2.46) in one cross-sectional study [32]. However, this result of a univariate analysis could not be confirmed in adjusted models, neither for men nor for women. Another cohort study defined this type of exposure as a risk factor for LE in men (incidence rate ratio (IRR) 2.80) [38]. This gender specific result in the cohort study adjusted for age and repetitiveness was only significant if information of baseline and follow-up investigation were implemented in the models.

Spahn et al. [40] reported that repetitions >3/s for >1 h/day (#8) could be a significant risk factor for LE in men (OR 10.60) and women (OR 11.00).

Nordander and colleagues [34] reported about elevated prevalence rates (significant for ME but not for LE) with increasing wrist angular velocity (risk factor #10).

Longer duty cycles (#54 [37]) or repetitive shoulder movements (#55, #56 [31]), on the other hand, showed no significant associations with LE.

#### 3.3.3. Posture/Movement

Spahn et al. [40] reported non-specific wrist extension >1 h/day as a significant risk factor for LE in men (OR 12.00) and women (OR 7.50), and non-specific wrist flexion >1 h/day was significantly associated with LE but only in men (OR 4.20). The authors also identified overhead working >1 h/day as a significant high risk for LE in men (OR 12.00), but not in women. Arm holding in front of the body or swinging movements of the arm >1 h/day (#66, #67) as well as general postures such as standing, sitting or PC work >1 h/day (risk factors #84 to #86) showed no significant associations in men and women.

A significant dose-response relationship was shown for frequently wrist bending or twisting for at least 2 h/day, which was associated with LE or ME (#13, #14, and #15). On the other hand, such bending or twisting for less than 2 h/day (#58) was not significantly associated with these disorders [36]. The same exposure for 2 to 4 h/day showed some significant associations with ME or LE/ME (OR 3.90 to 4.90), but not for LE only. Stronger associations occurred for wrist bending >4 h/day (#14) and the development of LE, ME or LE/ME (OR 4.40 to 8.20). All these results were based on univariate analyses. Furthermore, the risk for developing LE or LE/ME was significantly more than doubled (OR 2.70) if daily work involved forearm rotating for more than 4 h (#22). For <4 h forearm rotating per day (#68, #69) no significant results could be detected. The authors were able to find significant associations for men and women with LE, ME or LE/ME (OR 2.50 to 3.60) using multivariate analyses.

In contrast, other authors identified wrist bending or elbow flexion/extension for >2 h/day as a risk factor for LE in men (OR 2.27 to 2.41) and in women (OR 1.98 to 2.65) supported by univariate analyses [32].

Nordander et al. [34] demonstrated the association of wrist flexion and the prevalence of LE and ME: Each increasing degree of the wrist flexion angle (start at −40.0°, risk factor #16) was significantly associated with a 0.3% increased prevalence ratio (PR) of LE but not for ME. For ME, a 1° increase in the wrist flexion angle (start at −20.0°, risk factor #17) correlated with a 0.08% increase in PR. However, such sex-adjusted analyses were not significant for LE or ME at 0° flexion angle (#62).

In one study, non-neutral posture of the elbow or wrist ≥2 h/day (#23) was significantly associated with UN (OR 1.82) [39]. In these partly adjusted analyses, however, the associations between non-neutral postures of elbow or wrist ≥1 to <2 h/day (#70) were not significant. (Non-neutral postures were defined by the authors [39] as elbow flexion >100°, or ≥near maximal pronation/supination or wrist deviation (>5° radial, >10° ulnar) or >15° palmar/dorsal flexion according to other literature [45,46]).

Higher hazard ratios (HR) for LE (HR 2.25 to 3.10) are related to forearm pronation ≥45° for ≥40% of working time and ≥10% time of a duty cycle, and rotation (supination or pronation) ≥45° for ≥45% of working time and ≥10% time of a duty cycle, respectively [37]. Seventeen further risk-factor specifications (#60, #61, #64, #65, #71 to #83), reported by these authors, about unfavorable wrist flexion/extension or forearm pronation, supination, or rotation over time did not exhibit significant associations in univariate analyses or in adjusted ones for age and gender.

Although forearm supination ≥45° for ≥5% of working time (#26) did not show any significant associations in the cohort study [37] but was mentioned as a further risk-factor specification for LE (OR 2.25) in a cross-sectional study [31]. The latter result was proofed by analyses adjusted for age, gender, and BMI. Various frequencies of shoulder movements (#55, #56) were not significant as well as wrist radial deviation <5° or ulnar deviation ≥20° for ≥4% of time (#59) [31].

#### 3.3.4. Vibration

Svendsen and colleagues [39] described hand-arm vibrations (HAV) with acceleration ≥3 m/s² for >1 h/day as a significant risk factor (#27) for UN (OR 2.19) based on a job exposure matrix. Their result was based on models adjusted for age, gender, and BMI. On the other hand, vibrations >0 to <1 h/day were not significant for UN in their study. For LE, there were non-significant associations with the use of vibrating hand tools >2 h (#88 [32]) or vibration stress >1 h/day (#89 [40]).

#### 3.3.5. Combined Exposures

Forearm supination ≥45° for more than 5% of the working time combined with forceful lifting (≥4.5 kg object weight, risk factor #33, #34 [31]) was found in several adjusted analyses to be a risk factor for LE (OR 2.98 to 3.65). More favorable postures or lower forces (#91) as well as the effect of forearm supination ≥45° ≥5% (duty cycle) or forceful lifting (≥4.5 kg) >0% of time (#92) did not show significant associations with LE. In addition, these authors reported dose-response relationships of forceful exertions (≥44.1 N or ≥4.5 kg, (#29 to #32)) as times per minute or as percent of a duty cycle with significant effects (OR 3.00 to 5.17), supported by adjustments for several confounders.

Forearm pronation ≥45° for ≥40% of working time combined with one additional factor (any power grip (#38), lifting ≥3% of working time (#39)) was significantly associated with LE (HR 2.50 to 2.80) [37]. For this purpose, age- and gender-adjusted analyses were used. Furthermore, forearm supination ≥45° for less than 5% of working time combined with any power grip (#36) or lifting ≥3% of time (#35) showed higher significant hazard ratios for LE (HR 2.09 to 2.89), as did forearm rotation ≥45° for more than 45% of working time combined with any power grip (HR 2.83, risk factor #37). These results were based on univariate analyses. Besides frequency of forceful exertions (≥44.1 N or ≥4.5 kg) ≥2 times/min (#90), the authors demonstrated 40 further risk-factor specifications (#93 to # 117, #119 to #133) as effects of combined forces, postures, or repetitions. However, these did not reach statistical significance, although different models were used.

Other authors reported about combined physical exposures (defined as mostly hard physical exertion, corresponding to a level equal or greater than 14 on the 6 to 20 BORG Scale [47], combined with elbow movements >2 h/day) as a risk factor for LE in men and women. Elbow movements were defined as elbow flexion/extension more than 2 h/day and wrist bending more than 2 h/day [32,38]. The associated risk increased with greater numbers of elbow movements in men (OR 3.78 to 5.27) but not in women in the cross-sectional study [32]. Non-significant associations were found as well for less hard physical exertion (BORG Score 6 to 13 [47]) or less than 2 elbow movements [32].

In the subsequent cohort study [38] significant results could be shown for men and women (IRR 3.20 to 3.30), but only if the data of baseline and follow-up investigation were implemented in age and combined physical work exposure adjusted analyses.

Repetitive tasks executed with maximum force >1 h/day (#28) were significantly associated via sex-adjusted analyses with LE in men (OR 14.70) and in women (OR 29.30), whereas forceful turning >1 h/day (#40) was only significantly associated with LE in men (OR 4.70) but not in women [40].

Walker-Bone et al. [35] identified through multivariate analyses a significant relationship between repetitive bending or straightening of the elbow >1 h/day (#44) and the development of LE (OR 2.50) and ME (OR 5.30) in men and in women. 

Nordander et al. [33] found a high PR for ME (4.00) associated with work-related repetitive movements (cycle time >30 s) or constrained postures (>50% of working time). This specification was defined by previous information [48] and the result was significant only in men (no adjustments reported). Other disorders such as LE, radial tunnel syndrome or pronator teres syndrome were not significantly associated with this risk-factor specification (#43) [33].

#### 3.3.6. Evidence of Sub-Categories of Exposure

The GRADE evidence for prognostic factors was performed for all significant and non-significant associations.

For this purpose, the sub-categories of exposure (S1 to S16) were rated and assigned to the 4 possible gradations of the evidence evaluation (according to [26]) as follows:*High evidence* (*n* = 7): S2 (Forceful exertion), S4 (Manual load handling), S6 (Repetitiveness), S8 (Hand movements), S10 (Non-neutral posture), S13 (Force and repetition), S14 (Posture and force)*Moderate evidence* (*n* = 1): S16 (Posture and repetition and force)*Low evidence* (*n* = 3): S5 (High repetition), S9 (Forearm and elbow movements), S12 (Hand–arm vibration)*Very low evidence* (*n* = 5): S1 (Maximum force), S3 (Hand as tool), S7 (Overhead work), S11 (Body posture), S15 (Repetition and posture)

All evidence ratings are presented in Table 5.

## 4. Discussion

### 4.1. Quality of the Included Studies

#### 4.1.1. Study Design

Ten studies met our criteria, whereby both Fan et al. [31,37] and Herquelot et al. [32,38] published a cross-sectional study as the baseline of a cohort study and the follow-up of the same cohort in another article, respectively.

The methodological quality score for the included studies ranged from 10 (medium) to 14 (high) on a scale from 0 to 18. The main reason for not achieving the highest quality standards was a lack of blinding the exposure investigators with respect to the outcome or vice versa. This phenomenon seems to be wide spread in the related field of research, as other authors also mentioned this lack of quality [27]. Although all studies generally distinguished between exposed and non-exposed cases, the decision criteria were sometimes described very unclearly. Missing information on distributions of age, gender, or sport/leisure time, for example, led to a lower quality score of individual studies. The quality score also decreased due to a lack of information on sample size justification or power description. However, this information is very important for the reliability of a study and in our opinion should always be reported. Our final study pool contains 30% prospective longitudinal studies (cohort studies), 20% case/triple CRS, and 50% studies with a cross-sectional design (*n* = 10, see Table 3). Thus, this study pool lacks longitudinal studies for more valid proof of the outlined associations. On the other hand, this seems to be a general problem, as longitudinal studies are more time and cost consuming than CSS. Van Rijn et al. [6] reported similar results in their review: 15% cohort studies, 15% CRS and 70% CSS (*n* = 13).

#### 4.1.2. Elbow Disorders and Outcome Assessment

We found a high homogeneity among the studies for assessment of outcome, since all studies showed the highest quality score for this criterion (physical examination including clinical noticeable reduction of movement, clinical check, imaging procedure results and diagnosis). While our key paper included only 3/13 studies with more than 50 cases per investigation [6], 7/10 studies fulfilled this criterion in our review, indicating that elbow disorders may have become more important in today’s working environment. Finally, the studies included in our review focused predominantly on the same elbow disorders as van Rijn et al. [6] did (LE, ME, and UN). They described LE and ME as the most common disorders at the elbow, followed by cubital tunnel syndrome, which corresponds to UN in our review. Further disorders such as radial tunnel syndrome or pronator teres syndrome were investigated only in one study [33] and showed only non-significant effects for one risk-factor specification (#43). One reason for this could be that radial tunnel syndrome is one of the rare specific elbow diseases (IRR men 2.97 (1.9, 4.1), IRR women 1.42 (0.7, 2.2) per 100,000 years [14,49]) and is generally not as common as LE (IRR men 1.0 (0.7, 1.3), IRR women 0.9 (0.6, 1.3) per 100 workers) [38].

#### 4.1.3. Exposure Assessment

A high-quality exposure assessment (score = “5”) could only be found for 3 of the included studies [31,34,37]. This assessment quality mainly refers to the validity of quantitative physical exposures, since exposure measures are considered to have the highest quality when they are based on direct measurements or biomechanical model calculation.

Fan et al. [31,37] used time-based posture analysis via software and video frames, time studies, and force gauges, both at baseline and after 3.5 years follow-up. Nordander and colleagues [34] applied biaxial flexible electro-goniometers to measure wrist postures and movements. Additionally, they used surface electromyography (EMG) to record muscular load of the right forearm extensors in a sub-sample [34].

With regard to the review by van Rijn and colleagues [6], who included only 1 study with a similar high-quality exposure assessment, this may reflect technical progress in measurement equipment, which has become more applicable for use in the field. Nevertheless, the majority of our included studies performed exposure assessments at a lower quality level based on self-reports [32,35,36,38,40], on task-related exposure classifications [33], and on exposure classifications via job exposure matrix (JEM) and expert rating [39]. 

From an economic point of view, surveys or self-reports are superior to measurements. However, by trying to gain specific and detailed information about the association of exposure and the risk of developing specific disorders, quality standards—both for exposure and outcome assessment—should be as high as possible to our opinion.

### 4.2. Significant and Non-Significant Risk Factors

Force, repetition, posture/movement, vibration, and combined exposures were identified as the main exposure categories. These main categories contain 44 significant risk-factor specifications for developing at least one SDE. 89 non-significant specifications were reported by 9/10 studies. All 133 risk-factors specifications in total were summarized to 16 sub-categories of exposure, which were used for evidence ratings (see Table 5). These categories were similar to those reported by other authors [14].

Although most of the significant results were adjusted for at least one confounder, the interpretation of results from univariate analyses must nevertheless be carried out cautiously. In these univariate analyses, the true associative effect of a significant risk-factor specification could be covered by confounders. Results from multivariate analyses (adjusted for confounders) therefore appear somewhat more valid and should be preferred to results from univariate analyses.

Melhorn et al. [14] described e.g., gender with insufficient evidence for the development of LE or ME. The increasing age, on the other hand, increases the risk of developing LE or ME. At UN, these authors even attributed a strong evidence to age and some evidence to gender. We have tried to take these aspects into account by using predominantly adjusted results, if they were available. Furthermore, adjusting the statistical analysis to one or two potential confounders (e.g., age or gender) indirectly implies the influence of age and gender on SDE. However only very few of the included studies directly investigated an interaction of work-related physical risk factors and factors such as age, gender, or others (such as job experience) on SDE. Confounders such as job types played a very minor role for our aim, because we focused on movement executions (including forces, repetitions, postures, vibrations, and combinations) and not on job titles or job types in general.

#### 4.2.1. Force

This main exposure category contains 15 risk-factor specifications divided into 4 sub-categories with significant associations for LE, ME, and UN. These results were based on 6/10 studies and are consistent with the findings of van Rijn et al. [6]. Two sub-categories were rated with high evidence (Forceful exertion, Manual load handling), the other two sub-categories only with low (Hand as tool) or very low (Maximum force) evidence values.

Subsequently, although in the sub-categories “Hand as tool” and “Maximum force” high ORs were reported, those high risks have to be interpreted with caution, since the source study [40] was only rated with an assessment score of “3” (on a scale from 1 to 5, i.e., based on self-reports) and medium overall study quality. Moreover, isolated force exertions without the influence of posture or movement are unlikely to occur at the workplace. Therefore, values based on objective measurements using procedures such as EMG or force gauges might provide more reliable quantitative information as shown for manual handling of loads [31]. More precise exposure definitions of the force exposure sub-categories with high evidence ratings might support this presumption.

#### 4.2.2. Repetition

Repetition was a significant risk factor in 50% of the included studies and was confirmed for LE, ME, and UN, in both men and women. Van Rijn et al. [6] also outlined repetitive movements as a risk factor for LE and ME, but not for UN.

In contrast to force, repetition features higher evidence values ranging from low (S5 High repetition) to high (S6 Repetitiveness).

The 4 corresponding significant risk-factor specifications ranged from overall descriptions such as repetitive tasks ≥4 h/day [32,38] to very specific, measurement-based variables regarding wrist angular velocity [34]. To our opinion, results for very fast repetitions with more than 3 events per second (#8 [40]), derived from self-reports or surveys, should be interpreted carefully as they might be accompanied by strong recall bias, especially since the number of cases is small for a generalizable statement. As measurement-based exposure assessment shows the highest quality score, the related values might be most valid to assess repetitive tasks.

Previous systematic reviews [4,6] regarding work-related disorders at the elbow did not detect similar risk factors such as wrist angular velocity with high-quality scoring.

#### 4.2.3. Posture/Movement

Work-related awkward postures or movements of the upper limbs were associated with increased risks for elbow disorders in 70% of our included studies. From the 133 presented risk-factor specifications 44 were attributed to this main category and were predominantly linked to LE or ME, both in men and women, and seldom to UN. In 3 out of 10 included studies we found high exposure quality scores. One of them used objective measurements (wrist flexion [34]) and the other two used video-based analysis of non-neutral forearm postures ([31,37], see Appendix A (Appendix A)). However, 10/15 of the significant and 13/29 non-significant risk-factor specifications were recorded via self-administrated questionnaires, interviews, or JEMs.

Melhorn et al. [14] reported some evidence for awkward postures and showed aspects for posture as an independent risk factor. The wide range of evidence in our 5 corresponding sub-categories of exposure (S7 to S11, evidence: very low to high) might be due to their “easy to observe” status compared to factors such as force, vibration, or repetition. As observational studies are wide spread, the posture/movement (e.g., overhead work, #12) might be overrated compared to factors that are difficult to observe and require complex measurements to be detected validly.

Moreover, only one study in our results described overhead work (>1 h/day) as associated with LE in men [40]. The authors stated their study might be limited because of small sample sizes, which may affect the investigated associations. These findings are consistent with the results by van Rijn and colleagues [6], as their review contains only 1/13 studies mentioning “overhead work” as associated with LE in men and women. The authors reported static postures were linked with specific elbow disorders. However, we cannot confirm this in our review.

#### 4.2.4. Vibration

Although various studies included HAV into their exposure assessment [32,39,40], only one study mentioned HAV (>1 h/day) as a significant risk factor (for UN [39]). However, this association should be interpreted carefully, as the exposure assessment in this study was based on job titles and JEMs, respectively (exposure score “1”). Similarly to the moderate evidence for HAV, other authors reported insufficient evidence in relation to vibration [14].

One study also dealt with vibrations and pinch or power grip [31]. However, no statistical analyses were performed by the authors, which could indicate any association between vibration and LE. Therefore, we did not include these specifications in our results.

Van Rijn et al. [6] controversially discussed the relationship between LE and work-related vibrations. They included 3/13 studies assessing vibrations by self-reports with dissenting results regarding the associated disorders.

This might be a further hint that vibrations, especially hand-arm vibrations, are difficult to estimate by pure interviews or self-reports and need to be measured in future work for a valid quantitative specification.

#### 4.2.5. Combined Exposure

In our key paper [6], only qualitative relationships between LE and combinations of either force, repetition, or posture were outlined in 2/13 studies, although one of them used electromyography (EMG) and video for exposure assessment. In contrast, 70% of our studies illustrated the effect of combined exposures for 17 significant and 45 non-significant individual risk-factor specifications. Significant associations were predominantly demonstrated for LE in men and women, and were sporadically detected for ME in both sexes, but not for UN. Two studies [31,40] presented quantitative measures for the combination of force and repetition. The application of continuous EMG or force measures seems to be a suitable approach to quantitatively determine the risk of specific exposure combinations. Two studies found a dose-response relationship between EMG and force and elbow disorders [31,37].

Contrary to this, other quantitative information about the combined effect of repetition and awkward postures or movements (e.g., #44) were collected by self-reports and do not provide such detailed results [35].

The 4 corresponding sub-categories show a wide spectrum of studies with evidence ratings from very low (S15 Repetition and posture) to moderate (S16 Posture and repetition and force) up to high (S13 Force and repetition; S14 Posture and force). In addition, more significant associations could be identified in combined exposures of force and posture or repetition (S13, S14) than in posture and repetition (S15). Melhorn and colleagues [14] showed similar evidence ratings for grouped qualitative information on different risk factors specifications. They found a strong evidence for the combination of force and repetition or force and posture, whereas the evidence for Posture and repetition was rather classified as insufficient. In our results, 15/17 significant associations (#28 to #42) include the combination of force and either repetition or posture. In 11 of these 17 risk-factor specifications (#29 to #39), the specifications were measured, and the results were largely supported by adjusted models. The evidence also increased once force was involved. Based on our analyses and the work of Melhorn et al. [14], we believe that force combined with one additional exposure (posture or repetition) could have a major impact on the development of SDEs. Therefore, we would attribute greater importance to the significant quantitative risk-factor specifications (#28 to #42) than to the factors #43 and #44. On the other hand, the high ORs for factor #28 (maximum forceful efforts of the hand and repetition >1 h/day) should not be weighted too heavily, as the number of cases supported by one study [40] is small here. Finally, the high portion of measurement-based risk-factor specifications may support our assumption of the increased application of technical devices in epidemiological studies over the last years.

### 4.3. Strengths and Limitations

Our systematic literature review excels in a mixed search strategy including established methods (e.g., PICO [11], PRISMA [11,12]), various literature databases, peer-reviewed grey literature, and manual searching. This conservative approach was combined with a quality assurance procedure adapting the GRADE method for rating evidence [26,27,28,29]. For methodological quality assessment, we modified a published scoring system for rating hip exposures and hip outcomes [22] to address the elbow.

We relied on the risk-factor definitions that were available in the original studies even if they have been very heterogenous. However, with the help of the exposure assessment score, we at least tried to classify the exposure determination qualitatively.

We reported quantitative information instead of using a pure qualitative approach. Moreover, we demonstrated the impact of combined physical exposures on the development of SDEs. Thus, our results may help to develop or evaluate elaborated risk assessment methods for the elbow.

Our study focused on work-related physical exposure and neglected other possible risk factors, such as psychosocial influences which should be mentioned as a limitation or at least as a potential factor to be considered when continuing research on work-related risk factors and elbow disorders.

Most of the studies in this review used different study designs and several types of exposure descriptions and showed a high heterogeneity with respect to the quality of exposure assessment. Therefore, a meta-analysis was not feasible.

In addition, our search was time-restricted to one decade (2007 to 2017), and we are aware of missing information because of our rigorous exclusion criteria (e.g., publication language or specific study designs).

The inclusion of both cross-sectional and longitudinal studies may have elicited the high heterogeneity of the reported risk estimates, such as OR, PR, HR, or IRR, and made it difficult to compare their results. On the other hand, the exclusion of any of these study types would have led to a lack of information about potential risk factors.

We have tried to reduce the publication bias by presenting significant and non-significant results. Furthermore, we intended to keep the bias low with the help of evidence and bias assessment. However, we are aware that recall, information, and publication bias of the individual included studies nevertheless could have affected our results in some way. The small number of included studies may limit the general applicability of our findings.

We have presented EMG as a measurement data-based method and equated it with a high score for exposure assessment. However, EMG could be limited as well. Reasons for this could be, among other things, cross talk, loosening of electrodes during a measurement, high amount of subcutaneous fat tissue (electrode-skin impedance) or differences in the applied electrode attachment [50,51]. Nevertheless, EMG seems to be an adequate method to measure some of the relevant exposures and was one of the few methods that indicated measured exposures in our included studies.

## 5. Conclusions

Our study filled a gap in quantitative information about the association between work-related physical risk factors and SDEs, as the latest systematic approach on this topic was in 2007. We focused on quantitative measures of risk factors and combinations of risk factors. We identified 133 risk-factor specifications (44 significant, 89 non-significant), grouped them into 5 main- and 16 sub-categories of exposure, and assessed their scientific evidence.

Within the 16 sub-categories we found evidence ratings from very low to high based on a few studies only.

This highlights the need for further research in this area especially addressing the potential dose-response relationship of work-related exposures and specific disorders of the elbow. In this context, we consider the work of Fan et al. [31,37] and Nordander et al. [34] as important examples for assessing work-related physical risk factor of SDEs. Such objective measures may help to better describe the dose-response relationships between risk factors and SDEs in the future.

However, our results may be the base for developing or evaluating elaborated risk assessment methods for the elbow and, thus, an important step in preventing work-related disorders at the elbow. Furthermore, we would like to encourage other researchers to apply objective measures of exposure assessment in epidemiological studies to create an objective database and to better understand the impact of physical risk factors on WRULDs in the future. Therefore, this approach may be valuable for use in future research on WRULDs.

## Figures and Tables

**Figure 1 ijerph-16-00130-f001:**
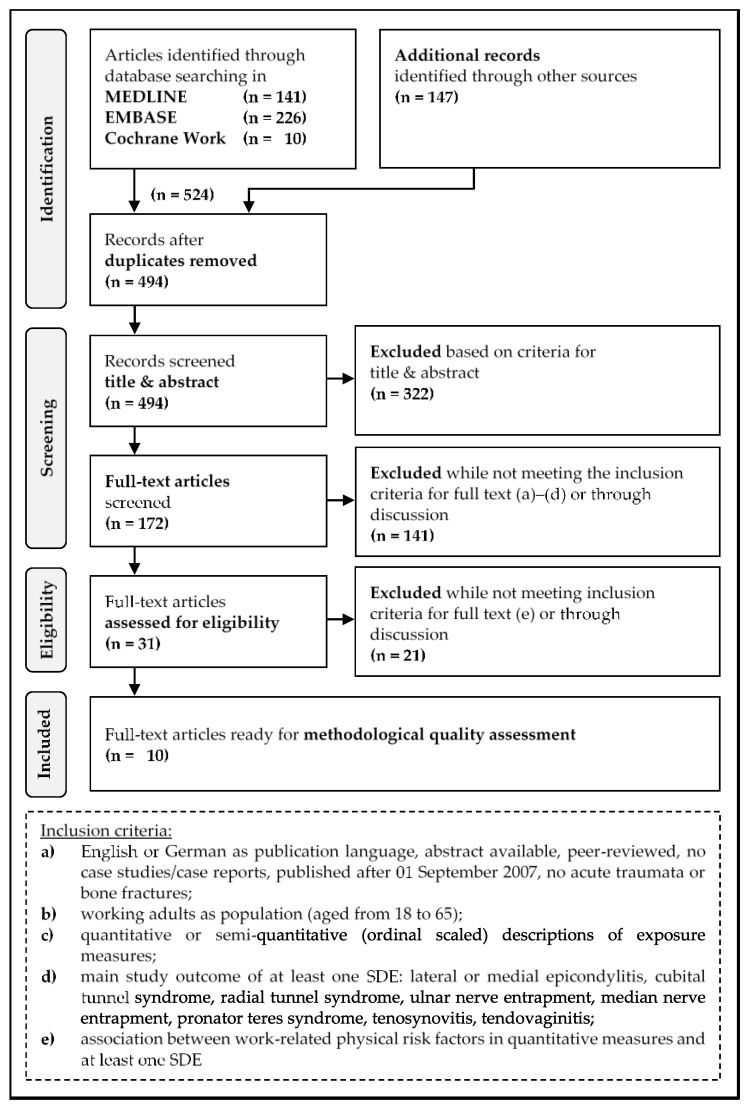
PRISMA Flow Diagram: This figure shows the study selection process of articles including physical risk factors associated with specific disorders at the elbow.

**Figure 2 ijerph-16-00130-f002:**
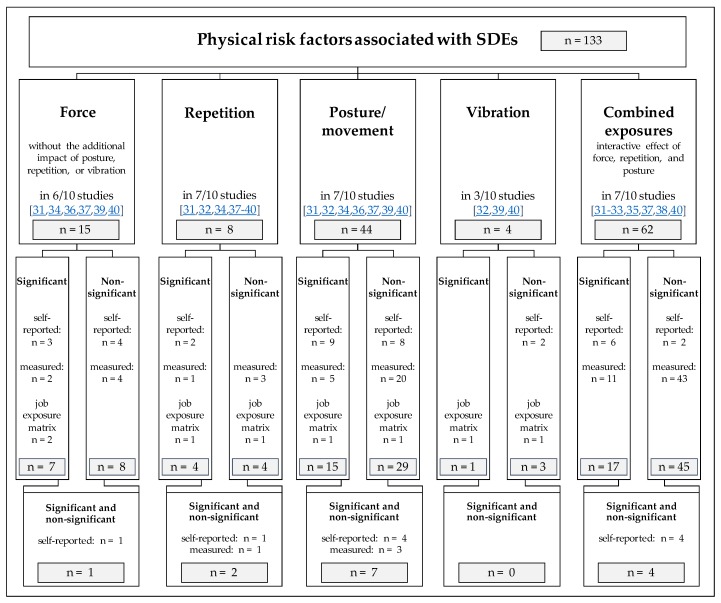
Distribution of physical risk factors associated with SDEs: The graph shows the distribution of all risk-factor specifications among the 5 main exposure groups with numbers of different exposure determinations and numbers of referring studies.

**Table 1 ijerph-16-00130-t001:** Applied general methodological quality assessment (study quality).

Criteria		Score
**Study population**		
1	Study groups are (exposed and unexposed) clearly defined	+/−/?
2	Participation ≥70%	+/−/?
3	Cases ≥50	+/−/?
**Assessment of exposure** (adequate description)		
4	Exposure definition	+/−/?
5	Assessment of exposure	+/−/?
6	Blind for outcome status	+/−/?
**Assessment of outcome** (specific disorder, adequate description)		
7	Outcome definition	+/−/?
8	Assessment method	+/−/?
9	Blind for exposure status	+/−/?
**Study design**		
10	Prospective design	+/−/?
11	Inclusion and exclusion criteria	+/−/?
12	Follow-up period ≥1 year	+/−/?
13	Information between completers vs. withdrawals	+/−/?
14	Research question *	+/−/?
**Analysis and data presentation**		
15	Data presentation identifying confounders	+/−/?
16	Consideration of confounders	+/−/?
17	Control for confounding	+/−/?
18	Statistical methods *	+/−/?

*Legend:* Item checklist for methodological quality assessment adopted from [6]; *Score building:* information about an item is either: available (+), not available (−), or unclear (?) based on the original study; *Symbols: ** Items added according to [19,20]; Item 14: Was the research question or objective in this paper clearly stated and appropriate? Item 18: Did the authors include a sample size justification, power description, or variance and effect estimates?

**Table 2 ijerph-16-00130-t002:** Applied quality assessment of exposure and outcome.

**Exposure Assessment**	**Score: Exposure ***
Profession, job title, classification of occupation	1
Qualitative specification of exposure in different work activities (standing, sitting, static or dynamic movements)	2
Quantitative specification of exposure in different work activities/physical strains with information on intensity (e.g., repetition, force, load weight, awkward postures, or duration)	3
Quantitative specification of exposure (as above) with additional plausibility check (e.g., information on daily work output or special controls through video analysis)	4
Direct measurement or biomechanical model calculation of elbow strain with specification of quantitative information (e.g., repetition per time, force, load weight, awkward postures, holding time of awkward postures, amount/amplitudes, acceleration, velocity, torque)	5
**Assessment of Outcome**	**Score: Diagnosis ****
Self-reported elbow pain without clinical check	1
Medical history/clinical questionnaire without clinical check or diagnosis	2
Clinically noticeable reduction of movement, clinical check, imaging procedure results and diagnosis	3

*Legend:* Score for exposure assessment: * Score 1 = low quality; Score 5 = high quality; Score for outcome assessment: ** Score 1 = low quality; Score 3 = high quality (modified according to [22]).

**Table 3 ijerph-16-00130-t003:** Quality assessment of included studies.

Reference	Study Design	Exposure Assessment Score *	Outcome Assessment Score *	General Methodological Quality Assessment (see Table 1 and chapter “Quality assessment” in ”Material and Methods” **)																			
				1	2	3	4	5	6	7	8	9	10	11	12	13	14	15	16	17	18	Score	Quality
Fan et al. (2009) [31]	CSS	5	3	+	+	−	+	+	+	+	+	−	+	+	+	−	+	+	+	+	−	14	high
Fan et al. (2014) [37]	CH	5	3	−	−	+	+	+	+	+	+	−	+	+	+	−	+	+	+	+	−	13	high
Herquelot et al. (2013b) [38]	CH	3	3	−	−	+	+	+	−	+	+	−	+	+	+	+	+	+	+	+	−	13	high
Descatha et al. (2013) [36]	CH	3	3	−	−	+	+	+	−	+	+	−	+	+	+	+	+	+	+	+	−	13	high
Svendsen et al. (2012) [39]	TCRS	1	3	−	−	+	+	+	−	+	+	−	−	+	−	+	+	+	+	+	+	12	high
Nordander et al. (2009) [33]	CSS	3	3	−	+	−	+	+	+	+	+	−	−	−	−	+	+	+	+	+	−	11	medium
Walker-Bone et al. (2012) [35]	CSS	3	3	−	−	+	+	+	−	+	+	−	−	+	−	−	+	+	+	+	−	10	medium
Herquelot et al. (2013a) [32]	CSS	3	3	−	−	+	+	+	−	+	+	−	−	+	−	−	+	+	+	+	−	10	medium
Nordander et al. (2013) [34]	CSS	5	3	−	+	−	+	+	+	+	+	−	−	−	−	−	+	+	+	+	−	10	medium
Spahn et al. (2016) [40]	CRS	3	3	+	+	+	−	−	−	+	+	−	−	+	−	−	+	+	+	+	−	10	medium
		Total item score		2	4	7	9	9	4	10	10	0	4	8	4	4	10	10	10	10	1	12	high

*Legend: Study design*: CSS = cross-sectional study; CH = cohort study; TCRS = triple case-referent study; CRS = case-referent study; *Symbols:* * Exposure Assessment Score: modified according to [22], max. quality score = “5”; Outcome Assessment Score: modified according to [22], max. quality score = “3”; descriptions and decision aids presented in Table 2; ** see Table 1 and Section 2.3; assessment according to [6], modified according to [19,20]; max. quality score = “18”; quality classification according to [19,21].

**Table 4 ijerph-16-00130-t004:** Overview of relevant physical risk factors significantly associated with the development of specific disorders at the elbow.

Exposure (Main- and Sub-Category (S) *)			Significant Risk-Factor Specification	Reference	Outcome	Gender	Measure (95%-CI)		Adjustment
Force	S1 Maximum force	#1	Maximum forceful efforts of the Hand >1 h/day	[40]	LE	Men	OR	6.90 (2.70, 17.50)	(g)
					LE	Women	OR	9.60 (3.10, 30.40)	(g)
	S2 Forceful exertion	#2	Hand in forceful grip on average ≥4 h/day	[36]	ME	-	OR	3.80 (1.50, 9.60)	(a)
					LE/ME	-	OR	2.80 (1.40, 5.80)	(a)
		#3	<10% maximum voluntary contraction across a full working day	[39]	UN	-	OR	2.73 (1.42, 5.25)	(h)
		#4	10 to 29% maximum voluntary contraction across a full working day	[39]	UN	-	OR	3.85 (2.04, 7.24)	(h)
	S3 Hand as tool	#5	Patting with the hand >1 h/day	[40]	LE	Men	OR	13.80 (2.90, 66.10)	(g)
	S4 Manual load handling	#6	Forceful lifting (≥4.5 kg) >0% of time	[31]	LE	-	OR	2.65 (1.21, 5.83)	(i)
		#7	Forceful lifting (≥4.5 kg) ≥2 times/min	[31]	LE	-	OR	3.06 (1.28, 7.27)	(i)
Repetition	S5 High repetition	#8	>3 motion sequences/sec or at least 10,000 times/h for >1 h/day	[40]	LE	Men	OR	10.60 (4.00, 28.30)	(g)
					LE	Women	OR	11.00 (2.60, 45.10)	(g)
	S6 Repetitiveness	#9	Doing repetitive tasks ≥4 h/day	[32]	LE	Women	OR	2.46 (1.30, 4.65)	(a)
				[38]	LE	Men	IRR	2.80 (1.20, 6.20)	(n)
		#10	Wrist angular velocity (5°/s) in [%/(°/s)]	[34]	ME	-	PR	0.10 (0.10, 0.20)	(g)
		#11	Repetitive elbow or wrist movements (≥4/min) ≥2.5 h/day	[39]	UN	-	OR	2.22 (1.41, 3.51)	(j)
Posture/movement	S7 Overhead work	#12	Overhead working >1 h/day	[40]	LE	Men	OR	12.00 (3.20, 43.80)	(g)
	S8 Hand movements	#13	Frequent wrist bending or twisting on average 2 to 4 h/day	[36]	ME	-	OR	4.90 (1.10, 20.70)	(a)
					LE/ME	-	OR	3.90 (1.10, 13.80)	(a)
		#14	Frequent wrist bending or twisting on average ≥4 h/day	[36]	LE	-	OR	4.40 (1.50, 13.10)	(a)
					ME	-	OR	8.20 (2.40, 27.90)	(a)
					LE/ME	-	OR	6.90 (2.40, 19.90)	(a)
		#15	Frequent wrist bending ≥4 h/day and forearm rotating on average ≥2 h/day	[36]	LE	-	OR	2.50 (1.10, 5.30)	(b)
					ME	-	OR	3.10 (1.40, 6.80)	(b)
					LE/ME	-	OR	3.00 (1.60, 5.80)	(b)
					LE/ME	Men	OR	2.80 (1.20, 6.20)	(b)
					LE/ME	Women	OR	3.60 (1.20, 11.00)	(b)
		#16	Wrist flexion (−40.0°) in [%/°]	[34]	LE	-	PR	0.30 (0.04, 0.60)	(g)
		#17	Wrist flexion (−20.0°) in [%/°]	[34]	ME	-	PR	0.08 (0.01, 0.10)	(g)
		#18	Wrist extension >1 h/day	[40]	LE	Men	OR	12.00 (3.00, 47.90)	(g)
					LE	Women	OR	7.50 (1.80, 31.60)	(g)
		#19	Wrist flexion >1 h/day	[40]	LE	Men	OR	4.20 (1.20, 14.80)	(g)
		#20	Extreme wrist bending >2 h/day	[32]	LE	Men	OR	2.27 (1.30, 3.97)	(a)
					LE	Women	OR	1.98 (1.04, 3.75)	(a)
	S9 Forearm and elbow movements	#21	Elbow flexion/extension >2 h/day	[32]	LE	Men	OR	2.41 (1.38, 4.22)	(a)
					LE	Women	OR	2.65 (1.40, 5.02)	(a)
		#22	Forearm rotating (also twisting, or screwing motion) ≥4 h/day	[36]	LE	-	OR	2.70 (1.20, 6.20)	(a)
					LE/ME	-	OR	2.70 (1.30, 5.40)	(a)
	S10 Non-neutral posture	#23	Non-neutral posture (elbow flexion >100°, or ≥near maximal pronation/supination; or wrist deviation (>5° radial, >10° ulnar) or >15° palmar/dorsal flexion) ≥2 h/day pronation/supination)	[39]	UN	-	OR	1.82 (1.15, 2.89)	(j)
		#24	Forearm rotation ≥45° for ≥45% time and duty cycle ≥10% of time	[37]	LE	-	HR	3.10 (1.05, 9.15)	(a)
		#25	Forearm pronation ≥45° for ≥40% time and duty cycle ≥10% of time	[37]	LE	-	HR	2.25 (1.09, 4.66)	(e)
		#26	Forearm supination ≥45° for ≥5% time	[31]	LE	-	OR	2.25 (1.13, 4.50)	(i)
Vibration	S12 Hand-arm vibration	#27	Hand-arm vibration: acceleration ≥3 m/s² ≥1 h/day	[39]	UN	-	OR	2.19 (1.05, 4.56)	(j)
Combined Exposures	S13 Force and repetition	#28	Maximum forceful efforts of the hand and repetition >1 h/day	[40]	LE	Men	OR	14.70 (5.20, 41.50)	(g)
					LE	Women	OR	29.30 (3.40, 34.80)	(g)
		#29	Frequency of forceful exertions (≥44.1 N or ≥4.5 kg) ≤1 to <5 times/min	[31]	LE	-	OR	4.47 (1.57, 13.71)	(d)
		#30	Frequency of forceful exertions (≥44.1 N or ≥4.5 kg) ≥5 times/min	[31]	LE	-	OR	5.17 (1.78, 15.02)	(d)
		#31	Duty cycle of forceful exertions (≥44.1 N or ≥4.5 kg) from ≤3 to <15% time	[31]	LE	-	OR	3.36 (1.28, 8.84)	(i)
		#32	Duty cycle of forceful exertions (≥44.1 N or ≥4.5 kg) for ≥15% time	[31]	LE	-	OR	3.00 (1.13, 7.96)	(i)
	S14 Posture and force	#33	Forearm supination ≥45° and forceful lifting (≥4.5 kg) in [% time]	[31]	LE	-	OR	3.65 (1.47, 9.07)	(i)
		#34	Forearm supination ≥45° ≥5% (duty cycle) and forceful lifting (≥4.5 kg) >0% of time	[31]	LE	-	OR	2.98 (1.18, 7.55)	(d)
		#35	Forearm supination ≥45° for <5% time and lifting (≥4.5 kg) ≥3% of time	[37]	LE	-	HR	2.09 (1.02, 4.27)	(a)
		#36	Forearm supination ≥45° for <5% time and any power grip (≥44.1 N)	[37]	LE	-	HR	2.86 (1.41, 5.82)	(a)
		#37	Forearm rotation ≥45° for ≥45% time and any power grip (≥44.1 N)	[37]	LE	-	HR	2.83 (1.16, 6.90)	(a)
		#38	Forearm pronation ≥45° for ≥40% time and any power grip (≥44.1 N)	[37]	LE	-	HR	2.80 (1.35, 5.77)	(e)
		#39	Forearm pronation ≥45° for ≥40% time and lifting (≥4.5 kg) ≥3% of time	[37]	LE	-	HR	2.50 (1.19, 5.24)	(e)
		#40	Forceful exertion (turning) >1 h/day	[40]	LE	Men	OR	4.70 (1.40, 16.20)	(g)
		#41	Hard physical exertion (BORG Score 14 to 20) and 1 elbow movement	[32]	LE	Men	OR	3.78 (1.85, 7.70)	(m)
		#42	Hard physical exertion (BORG Score 14 to 20) and 2 elbow movements	[32]	LE	Men	OR	5.27 (1.93, 14.37)	(a)
			(elbow movements = elbow flexion/extension >2 h/day and wrist bending >2 h/day						
			High physical exertion with elbow flexion/extension >2 h/day and extreme wrist bending >2 h/day (at follow-up investigation)	[38]	LE	Men	IRR	2.70 (1.10, 6.10)	(f)
			High physical exertion with elbow flexion/extension >2 h/day and extreme	[38]	LE	Men	IRR	3.20 (1.50, 6.40)	(f)
			wrist bending >2 h/day (at baseline and at follow-up investigation)		LE	Women	IRR	3.30 (1.40, 7.60)	(f)
	S15 Repetition and posture	#43	Repetitive/constrained work with >30 s or >50% of cycle time (involved same	[33]	ME	Men	PR	4.00 (1.10, 15.00)	(c)
			fundamental cycle) vs. >50% (working time) involved prolonged awkward postures						
		#44	Repetitive bending/straightening of the elbow >1 h/day	[35]	LE	-	OR	2.50 (1.20, 5.30)	(k)
					ME	-	OR	5.30 (1.90, 14.90)	(k)

*Legend:* * Forceful exertions were only shown for power grip, details for pinch grip were shown in Appendix A (Appendix A); sub-categories of exposure S11 and S16 are not significant and are not presented here. *Outcome:* UN = ulnar neuropathy; LE = lateral epicondylitis; ME = medial epicondylitis; LE/ME = lateral and/or medial epicondylitis; *Measure:* odds ratio (OR); hazard ratio (HR); incidence rate ratio (IRR); prevalence ratio (PR); *Adjustment:* (a) = univariate analysis; (b) = multivariate analysis; adjustment not reported; (c) = adjustment not reported; (d) = final model (age, gender, body mass index (BMI), smoking status, personal, psychosocial, and work organizational variables); (e) = adjusted for age and gender; (f) = adjusted for age and combined physical work exposure including physical exertion and elbow movements; (g) = sex-adjusted; (h) = fully adjusted for body mass index, pack-years of smoking (continuous), alcohol consumption (continuous), side-specific fractures (never/ever), full anesthesia within a 5-year period up to the index year (no/yes), predisposing disorders (no/yes), use of crutches within a 5-year period up to the index year (no/yes), hand-arm intensive sports (0, 1, 2) and weight loss ≥10 kg within half a year during a 5-year period up to the index year (no/yes) and all occupational exposure variables in Table 2 of [39]; (i) = adjusted for age (continuous), gender, BMI (continuous); (j) = partly adjusted for body mass index, pack-years of smoking (continuous), alcohol consumption (continuous), side-specific fractures (never/ever), full anesthesia within a 5-year period up to the index year (no/yes), predisposing disorders (no/yes), use of crutches within a 5-year period up to the index year (no/yes), hand–arm intensive sports (0, 1, 2) and weight loss ≥10 kg within half a year during a 5-year period up to the index year (no/yes); (k) = multivariate analyses; adjusted for vitality, white/blue collar, age in four age bands and sex; (m) = adjusted for individual characteristics, repetition, combined physical work exposure including physical exertion, elbow flexion/extension and wrist bending, and social support with aggregation of low categories for combined physical work exposure; (n) = adjusted for age and repetitiveness.

**Table 5 ijerph-16-00130-t005:** Rating evidence by using Grading of Recommendations, Assessment, Development and Evaluation (GRADE) according to [26,27].

Exposure (Main- and Sub-Category (S))	Risk Factors			*n*	Number of Studies	Number of Cohorts	Outcome	Uni-/Multivariate Analyses						GRADE Factors (According to [26,27])								GRADE Evidence
								+	0	−	+	0	−	I	II *	III	IV	V	VI **	VII	VIII	
**Force**	S1	Maximum force	#1	197	1 [40]	0	LE				2	0	0	1	2↓	n. a.	1↓	1↓	✔	↑	†	Very low
	S2	Forceful exertion	#2 to #4, #46 to #49	8055	3 [34,36,39]	1	LE; ME LE/ME; UN	2	7	0	2	1	2	2	2↓	✔	✔	✔	1↓	↑	↑	High
	S3	Hand as tool	#5	197	1 [40]	0	LE				1	0	0	1	2↓	n. a.	1↓	1↓	✔	↑	†	Very low
	S4	Manual load handling	#6, #7, #50 to #53	930	3 [31,37,40]	1	LE	0	1	0	2	3	0	3	1✔	✔	✔	1↓	✔	↑	↑	High
**Rep-etition**	S5	High repetition	#8	197	1 [40]	0	LE				2	0	0	1	2↓	n. a.	✔	1↓	✔	↑	†	Low
	S6	Repetitiveness	#9 to #11, #54 to #57	11391	6 [31,32,34,37,38,39]	2	LE; ME; UN	1	1	0	3	11	0	3	1✔	1↓	✔	✔	✔	†	†	High
**Posture/Movement**	S7	Overhead work	#12	197	1 [40]	0	LE				1	1	0	1	2↓	n. a.	1↓	1↓	✔	↑	†	Very low
	S8	Hand movements	#13 to #20, #58 to 63	8399	6 [31,32,34,36,37,40]	2	LE; ME; LE/ME	7	5	0	10	7	0	3	2↓	✔	✔	1↓	1↓	↑	↑	High
	S9	Forearm and elbow movements	#21, #22, #66 to #69	5014	3 [32,36,40]	1	LE; ME LE/ME	4	7	0	0	4	0	2	2↓	✔	✔	1↓	1↓	↑	†	Low
	S10	Non-neutral posture	#23 to #26, #64, #65, #70 to #83	5029	3 [31,37,39]	1	LE; UN	1	14	0	3	3	0	3	1✔	✔	✔	1↓	✔	†	†	High
	S11	Body posture	#84 to #86	197	1 [40]	0	LE				0	6	0	1	2↓	n. a.	1↓	1↓	✔	†	†	Very low
**Vibration**	S12	Hand–arm vibration	#27, #87 to #89	8203	3 [32,39,40]	0	LE; UN	0	2	0	1	2	0	1	2↓	✔	✔	✔	✔	†	†	Low
**Combined exposure**	S13	Force and repetition	#28 to #32, #90	930	3 [31,37,40]	1	LE	0	1	0	6	0	0	3	1✔	✔	✔	1↓	✔	↑	↑	High
	S14	Posture and force	#33 to #42, #45, #91 to #118	4640	5 [31,32,37,38,40]	2	LE	4	22	0	9	15	0	3	1✔	1↓	✔	1↓	✔	↑	↑	High
	S15	Repetition and posture	#43, #44	8690	2 [33,35]	0	LE; ME; Pronator; Radial				3	6	0	1	2↓	1↓	✔	1↓	✔	↑	†	Very low
	S16	Posture and repetition and force	#119 to #133	611	1 [37]	1	LE	0	15	0				3	1✔	n. a.	✔	1↓	1↓	†	†	Moderate

*Legend:* table layout, column descriptions according to different authors [26,27]; *Shortcuts: n* = number of participants; *Outcome:* UN = ulnar neuropathy; LE = lateral epicondylitis; ME = medial epicondylitis, LE/ME = lateral and/or medial epicondylitis, Radial = Radial tunnel syndrome, Pronator = Pronator teres syndrome; *Uni-/Multivariate Analyses:* type of analysis univariate/multivariate; + = number of significant effects with a positive value; 0 = number of non-significant effects; **−** = number of significant effects with a negative value; *GRADE factors:* I = phase of investigation; II = study limitations; III = inconsistency; IV = indirectness; V = imprecision; VI = publication bias; VII = moderate/large effect size; VIII = dose effect; *Symbols:* ↓ significant downgrading; ✔ no serious limitations/no downgrading); n. a. = not applicable; ↑ significant upgrading; † no upgrading; * Bias assessment according to [28,29]; ** overall quality of evidence was downgraded by phase of investigation: no downgrading.

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
