# Peer review of "Quantitative Measures of Physical Risk Factors Associated with Work-Related Musculoskeletal Disorders of the Elbow: A Systematic Review"

_ijerph, 2019, doi:10.3390/ijerph16010130_

Round 1

Reviewer 1 Report

Summary

This study presented a review results on evidence-based quantitative measure of physical risk factors associated with specific elbow disorder. The study identified a total of 133 different risk factors from 10 articles over the 10 years period of study. The significant risk factors were found to be the wrist angular velocity and forearm supination which are associated with lateral and medial epicondylitis, or ulnar neuropathy.

General Comments

The study was well executed and summarized interesting result. The manuscript was also well written. However, there are some sentence correction and proofreading are required. In addition, the manuscript is little lengthy. Specifically, the result section should be presented with figures and tables instead of a lot of written description.

Specific Comments

Introduction

Line 38: Upper limb disorders include shoulder, elbow, and wrist joint. Among them, what percent are specifically for elbow joint?  

Results

Line 168: Sentence doesn’t seem complete and clear

The results section is too long and heavy wordy. First, two paragraphs should be in the method section. There is no figure in the result section. Instead of explaining in sentences, the author should use some figures to explain the results. 

Author Response

Dear reviewer, dear editors,

Thank you very much for the quick review process. We have modified the manuscript a little using the reviewer's notes. Corrections or inserted sentences were made in Word edit mode. I.e. these would have to be fixed by clicking on Accept suggestions. However, inserted bibliographical references are not in edit mode, since from my experience sometimes such links could be changed or destroyed by the edit mode.

The implementation of the reviewer notes in the manuscript can be described as follows:

Reviewer 1 - General Comments

 The study was well executed and summarized interesting result. The manuscript was also well written. However, there are some sentence correction and proofreading are required.

We read the manuscript again, checking it out, and added small corrections (like missing characters, letters, grammar, etc.) in edit mode.

In addition, the manuscript is little lengthy. Specifically, the result section should be presented with figures and tables instead of a lot of written description.

We have tried to loosen this up by correcting the compact text of the results and moving the existing graphics and tables to other places in the text. Furthermore, we have shortened the text in the results section as desired and inserted a new graphic (Figure 2).

 Specific Comments

Introduction

Line 38: Upper limb disorders include shoulder, elbow, and wrist joint. Among them, what percent are specifically for elbow joint?  

The missing information was added in one sentence.

Results

Line 168: Sentence doesn’t seem complete and clear

Thank you. This sentence has been rewritten and divided into 2 sentences.

The results section is too long and heavy wordy. First, two paragraphs should be in the method section.

The chapter on methodological quality in the results section was divided into 2 subchapters.

There is no figure in the result section. Instead of explaining in sentences, the author should use some figures to explain the results. 

The existing graphic (Figure A1) has been moved from the Appendix to the main text for better readability. All text passages referring to Figure A1 were adapted as normal Figure 1. An other Figure (Figure 2) was created and inserted instead of some written text.

Is a better quality of Figure 2 (*.tiff) and a manuscript version without edit/ tracking mode required?

Please do not hesitate to contact me if you have any questions about our modification or the word edit mode.

Thank you 

Kind regards

David Seidel (on behalf of all the authors)

Reviewer 2 Report

The systematic review is well written and well organised. 

Some specific comments: 

I would suggest to add 1 table to show the study designs, subject groups (job types), outcome measures etc of the included studies. 

In the results and discussion, there is no mention of the effects of demographics (age and gender), job types of the subjects in various studies, 

In the conclusion, you should highlight which factors are found to be important and any recommendations from this systematic review? 

If these can be addressed, the paper can be published. 

Author Response

Dear reviewer, dear editors,

Thank you very much for the quick review process. We have modified the manuscript a little using the reviewer's notes. Corrections or inserted sentences were made in Word edit mode. I.e. these would have to be fixed by clicking on Accept suggestions. However, inserted bibliographical references are not in edit mode, since from my experience sometimes such links could be changed or destroyed by the edit mode.

The implementation of the reviewer notes in the manuscript can be described as follows:

We read the manuscript again, checking it out, and added small corrections (like missing characters, letters, grammar, etc.) in edit mode.

Reviewer 2: Some specific comments: 

I would suggest to add 1 table to show the study designs, subject groups (job types), outcome measures etc. of the included studies. 

This information was also very important for us. Therefore, the information on outcome assessment, outcome, study design, etc. was already listed in a table. This was in the Supplementary File VII; Table S5. Therefore, we did not duplicate the information by inserting an additional table here.

In the results and discussion, there is no mention of the effects of demographics (age and gender), job types of the subjects in various studies, 

We have included this aspect. It can be found as an inserted text passages in chapter 3.3. “Physical risk factors associated with SDEs” and in chapter 4.2. "Significant and non-significant risk factors".

In the conclusion, you should highlight which factors are found to be important and any recommendations from this systematic review? 

We have taken this suggestion into account and inserted some sentences in the Conclusion section.

By moving graphics or tables to another place in the manuscript for better readability of the text, we have updated the supplementary file again, so that the references to page numbers of the manuscript are correct again.

Is a manuscript version without edit/ tracking mode required?

Please do not hesitate to contact me if you have any questions about our modification or the word edit mode.

Thank you 

Kind regards

David Seidel (on behalf of all the authors)